# FEW-SHOT REGRESSION VIA LEARNED BASIS FUNCTIONS

**Yi Loo**[1]**, Swee Kiat Lim**[2]**, Gemma Roig**[1]**, Ngai-Man Cheung**[1]
[1] ST Engineering Electronics - SUTD Cyber Security Laboratory,
Singapore University of Technology and Design
[2] Singapore University of Technology and Design
{loo_yi,ngaiman_cheung,gemma_roig}@sutd.edu.sg,
sweekiat_lim@mymail.sutd.sdu.sg

## ABSTRACT

The recent rise in popularity of few-shot learning algorithms has enabled models to quickly adapt to new tasks based on only a few training samples. Previous few-shot learning works have mainly focused on classification and reinforcement learning. In this paper, we propose a few-shot meta learning system that focuses exclusively on regression tasks. Our model is based on the idea that the degree of freedom of the unknown function can be significantly reduced if it is represented as a linear combination of a set of appropriate *basis functions*. This enables a few labelled samples to approximate the function. We design a Feature Extractor network to encode basis functions for a task distribution, and a Weights Generator to generate the weight vector for a novel task. We show that our model outperforms the current state of the art meta-learning methods in various regression tasks.

## 1 INTRODUCTION

Regression deals with the problem of learning a model relating a set of inputs to a set of outputs. The learned model can be thought as function $\boldsymbol{y} = F(\boldsymbol{x})$ that will give a prediction $\boldsymbol{y} \in \mathbb{R}^{d_y}$ given input $\boldsymbol{x} \in \mathbb{R}^{d_x}$ where $d_y$ and $d_x$ are dimensions of the output and input respectively. Typically, a regression model is trained on a large number of data points to be able to provide accurate predictions of new inputs. Recently, there have been a surge in popularity on *few-shot learning* methods (Vinyals et al., 2016; Koch et al., 2015; Gidaris & Komodakis, 2018). Few-shot learning methods require only a few examples from each task to be able to quickly adapt and perform well on a new task. The few-shot learning model in essence is *learning to learn* i.e. the model learns to quickly adapt itself to new tasks rather than just learning to give the correct prediction for a particular input sample.

In this work, we propose a few shot learning model that targets few-shot regression tasks. We evaluate our model on the sinusoidal regression tasks and compare our model's performance to several meta-learning algorithms. We further introduce two more regression tasks, namely the 1D heat equation task modeled by partial differential equations and the 2D Gaussian distribution task.

## 2 RELATED WORK

Regression problems has long been a topic of study in the machine learning and signal processing community (Myers & Myers, 1990; Specht, 1991). Though similar to classification, regression estimates one or multiple scalar values and is usually thought of as a single task problem. A single model is trained to only perform regression on a only one task. Our model instead reformulates the regression problem as a few-shot learning problem, allowing for our model to be able to perform regressions of tasks sampled from the same task distribution.

The problem of meta-learning has similarly long been a topic of interest in the general machine learning community (Thrun & Pratt, 1998; Schmidhuber, 1987; Naik & Mammone, 1992). Meta learning has been applied to few-shot learning problem, which is concerned with models that can learn from prior experiences to adapt to new tasks. Lake et al. (2011) first proposed the one-shot

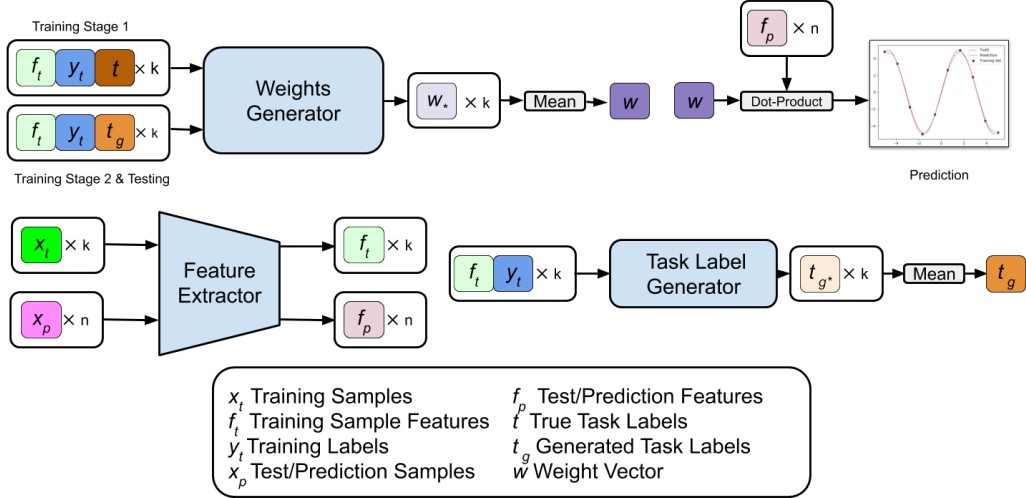

Figure 1: An overview of our model. Note that during meta-training, we use the true task labels of the regression task as input to the Weights Generator to train both the Weights Generator and Feature Extractor, whereas the generated task labels from the Task Label Generator are only used during meta-testing.

classification problem in 2011 and introduced the Omniglot data set as a few-shot classification data set similar to how the MNIST data set (LeCun, 1998) is for traditional classification. Since then, there has been a surge of few-shot learning methods (Vinyals et al., 2016; Finn et al., 2017; Gidaris & Komodakis, 2018; Rusu et al., 2018), but most of them focus on few shot classification and reinforcement learning domains.

## 3    FEW-SHOT REGRESSION VIA LEARNED BASIS FUNCTIONS

We first discuss our idea. We aim at developing a model that can rapidly adapt to regress a *novel* function based on only a few samples from this function. Specifically, we would like to model the unknown function $\boldsymbol{y} = F(\boldsymbol{x})$ given only $\mathbb{D}_{train} = \{(\boldsymbol{x}^k, \boldsymbol{y}^k)|k = 1...K\}$. With small $K$, e.g. $K = 10$, this is an intractable task, as $F(\boldsymbol{x})$ can take any form. We follow the common setup: we assume that each function we would like to regress is a task $\mathcal{T}_i$ drawn from an *unknown* distribution $p(\mathcal{T})$.

To simplify discussion, we assume scalar input and scalar output. Our idea is to learn *sparse* or *compressible* representation of the unknown function $F(x)$, so that a few samples $\{(x^k, y^k)|k = 1...K\}$ can provide adequate information to estimate $F(x)$. Specifically, we model the unknown function $F(x)$ as a linear combination of a set of *basis functions* $\{\phi_i(x)\}$:

$$F(x) = \sum_i w_i \phi_i(x) \tag{1}$$

Many basis functions have been developed to expand $F(x)$. For example, the Maclaurin series expansion (Taylor series expansion at $x = 0$) uses $\{\phi_i(x)\} = \{1, x, x^2, x^3, ...\}$:

$$F(x) = w_0 + w_1 x + w_2 x^2 + ... \tag{2}$$

If $F(x)$ is a polynomial, (2) can be a compressible representation, i.e. only a few non-zero/significant $w_i$. However, if $F(x)$ is a sinusoid, it would require many terms to represent $F(x)$ adequately, e.g.:

$$\sin(x) \approx w_1 x + w_3 x^3 + w_5 x^5 + w_7 x^7 + ... + w_M x^M \tag{3}$$

In (3), $M$ is large and $M \gg K$. Given only $K$ samples $\{(x^k, y^k)|k = 1...K\}$, it is not adequate to determine $\{w_i\}$ and model the unknown function. On the other hand, if we use the Fourier basis instead, i.e., $\{\phi_i(x)\} = \{1, \sin(x), \sin(2x), ..., \cos(x), \cos(2x), ...\}$, clearly, we can obtain a sparse

representation: we can represent a sinusoid with only a few terms. Under Fourier basis, there are only a few non-zero linear weights $w_i$, and $K$ samples are sufficient to estimate $w_i$ and estimate the function. **Essentially, with an appropriate $\{\phi_i(x)\}$, the degree of freedom of $F(x)$ can be significantly reduced when it is modeled using (1), so that $K$ samples can well estimate $F(x)$.**

Our approach is to use the set of training tasks drawn from $p(\mathcal{T})$ to learn $\{\phi_i(x)\}$ that result in sparse/compressible representation for any task drawn from $p(\mathcal{T})$. The set of $\{\phi_i(x)\}$ is encoded in the **Feature Extractor** that takes in $x$ and outputs $\Phi(x) = [\phi_1(x), \phi_2(x), ..., \phi_M(x)]^T$. In our framework, $\Phi(x)$ is the same for any task drawn from $p(\mathcal{T})$, as it encodes the set of $\{\phi_i(x)\}$ that can sparsely represent any task from $p(\mathcal{T})$. We further learn a **Weights Generator** to map the $K$ training samples of a novel task to a constant vector $\boldsymbol{w} = [w_1, w_2, ..., w_M]^T$. The unknown function is modeled as $\boldsymbol{w}^T \Phi(x)$.

### 3.1 MODEL ARCHITECTURE

We hereby introduce our few-shot regression model in detail. Given a regression task $\mathcal{T}$ with $\mathbb{D}_{train} = \{(\boldsymbol{x}^k, \boldsymbol{y}^k)|k = 1...K\}$, the model is tasked to predict the entire regression function across a value range.

The training samples, $\boldsymbol{x} \in \mathbb{R}^{d_x}$ first passed though the **Feature Extractor** which is represented as a function, $F(\boldsymbol{x}|\theta^F)$ with trainable parameters $\theta^F$. The Feature Extractor outputs a high dimensional feature representation, $\boldsymbol{f} \in \mathbb{R}^{d_f}$, where $d_f$ is the dimension of the feature representation, for each training point of the task $\mathcal{T}$. Note that $d_f$ is the number of basis functions encoded in the Feature Extractor.

The feature representation $\boldsymbol{x}_f$, together with the labels $y \in \mathbb{R}^{d_y}$ and task labels $\boldsymbol{t} \in \mathbb{R}^{d_t}$ generated from the Task Label Generator are then passed through the **Weights Generator**. The Weights Generator, represented as a function $G(\boldsymbol{f}, \boldsymbol{y}, \boldsymbol{t}|\theta^G)$, with trainable parameters $\theta^G$, outputs a weights vector, $\boldsymbol{w}_k$ for each training sample of a regression task. The final weights vector, $\boldsymbol{w}'$ for task $\mathcal{T}$ is then obtained by taking a mean of the $k$ weight vectors. The Weights Generator itself consists a series self-attention blocks with scaled dot product attention introduced by Vaswani et al. (2017). Each of the self-attention modules allows the weights generator to "look" at the embedding of the Weights Generator's input to let generator "choose" the parts of the embedding which is most useful in generating the optimal weights for each of the training sample.

The model is then able to make predictions on set of points $\mathbb{D}_{pred} = \{\boldsymbol{x}^n|n = 1...N\}$ for task $\mathcal{T}$, by taking a dot product between task-specific weights vector, $\boldsymbol{w}$ and feature representation of the prediction set.

$$\boldsymbol{y}' = \boldsymbol{w}^T F(\boldsymbol{x} \in \mathbb{D}_{pred}|\theta^F) \tag{4}$$

Note that for all of our regression experiments, $\boldsymbol{y}$ has a dimension of 1. However, our model is able to predict regression tasks with higher dimensional $\boldsymbol{y}$ by outputting $d_y$ weight vectors from the Weights Generator, the predictions can be obtained by doing a dot product at each dimension of $\boldsymbol{y}$ between the individual weight vectors and $\boldsymbol{f}$. An overview of our model can be found in Figure 1.

### 3.2 THE TASK LABEL GENERATOR

Outside of the label information $\boldsymbol{y}$ at the sample level, few-shot regression tasks, unlike other few-shot learning tasks also possess additional label information at the task level. These task level labels are parameters that describe a regression function that we can leverage to improve the performance of a few-shot regression model. For example, a sinusoidal function has parameters labels such as the amplitude, phase and frequency. We dub these task-level labels as task labels, $\boldsymbol{t}$ and we use it as an additional input to the Weights Generator.

Though we assume that the model to have access to task labels during the training phase, it is unrealistic to assume that such information will be available or reliable as well during testing. Therefore, we introduce the Task Label Generator as an additional component to our model. We represent it as a function $T(\boldsymbol{x}_f, y|\theta^T)$ with trainable parameters $\theta^T$. It takes in feature representation $\boldsymbol{x}_f$ and labels $\boldsymbol{y}$ of the regression task $\mathcal{T}$ and attempts to output the correct task labels, $\boldsymbol{t}_g$ for $\mathcal{T}$. Similar to the

Table 1: Comparing Mean-Squared Error between MAML and Our Model on different regression tasks. Lower is better.

| Regression Data | Meta-SGD (Li et al., 2017) | Ours with Task Label Generator | |
| --- | --- | --- | --- |
| Sinusoid | $0.53 \pm 0.09$ | $\mathbf{0.465 \pm 0.022}$ | |
| **Regression Data** | **BMAML (Yoon et al., 2018)** | **Ours with pre-trained Task Label G.** | |
| Sinusoid w/ noise | $\sim 1.35$ | $\mathbf{1.31 \pm 0.05}$ | |
| **Regression Data** | **MAML (Finn et al., 2017)** | **Ours w/o Task Labels** | **Ours with Task Label G.** |
| 1D Heat Equation | $0.0257$ | $0.00242 \pm 0.00008$ | $\mathbf{0.00131 \pm 0.00006}$ |
| 2D Gaussian | $0.0220$ | $\mathbf{0.00413 \pm 0.00209}$ | $0.00449 \pm 0.00328$ |

Table 2: Experiment on the effects of adding task labels.

| Regression Data | Using True Task Labels | No Task Labels | w/ Task Label Generator |
| --- | --- | --- | --- |
| Sinusoid | $0.0187 \pm 0.00039$ | $0.743 \pm 0.032$ | $0.465 \pm 0.022$ |

Weights Generator, we also employ the use of self-attention blocks within the Task Label Generator to enable the Task Label Generator to "look" at parts of the input that are most useful to generating the correct task labels.

## 4 RESULTS AND EVALUATION

We evaluate our model on three few-shot regression tasks. The first task is the sinusoidal regression task which has been used in many recent few-shot learning papers (Finn et al., 2017; Rusu et al., 2018; Li et al., 2017). We also introduce two more regression tasks, namely the 1D heat equation task (Cannon, 1984) and the higher dimensional 2D Gaussian distribution task. For the sinusoidal task we compare our model to Meta-SGD proposed by Li et al. (2017). We also compare our results on sinusoidal task to that of Yoon et al. (2018) where they add a noise component and only train with limited number of tasks for added complexity. We follow their set up of training of model with only 1000 tasks but we used a pre-trained Task Label Generator instead. For 1D Heat Equation and 2D Gaussian, we compare our models performance against MAML (Finn et al. (2017) and include results of our model with and without using the Task Label Generator.

We calculate mean squared error across 1000 test tasks for all regression tasks and present our results in Table 1. Our model manages to outperform both Meta-SGD and BMAML on both variants of the sinusoidal task. Our model also manage to achieve a superior performance on 1D Heat Equation and 2D Gaussian. Our results show that even without the use of task labels, our model setup is already able to perform well in the two regression tasks.

### 4.1 ABLATION STUDY

Furthermore, we conduct a ablation study to study the effects of adding the Task Label Generator. We compare three variants of our model an evaluate them on the advanced sinusoidal task .For the first variant when we use the true task labels in both training and testing. In the second variant, we do not use task labels at all. Finally we compare the two variants to the base model. We show the results of this study in Table 2.

## 5 CONCLUSION

We propose a few-shot meta learning system that focuses exclusively on regression tasks. Our model is based on the idea of linear representation of basis functions. We design a Feature extractor network to encode the basis functions for the entire task distribution. We design a Weight generator network to generate the weights from the $K$ training samples of a novel task drawn from the same task distribution. We show that our model has competitive performance in in various few short regression tasks.

ACKNOWLEDGMENTS

This work was supported by both ST Electronics and the National Research Foundation (NRF), Prime Minister's Office, Singapore under Corporate Laboratory @ University Scheme (Programme Title: STEE Infosec - SUTD Corporate Laboratory). The authors would also like to thank the anonymous reviewers for their valuable comments.

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

# A   TECHNICAL DETAILS

## A.1   THE FEATURE EXTRACTOR

We represent the feature extractor as a a 2 layer 40-dimensional fully connected network which transforms each 1-dimensional input sample into a 40-dimensional feature, Each hidden layer in the feature extractor also followed by a ReLU non-linearity activation function (Nair & Hinton, 2010).

## A.2   THE WEIGHTS GENERATOR

The Weights Generator takes in a $41 + t_n$ dimensional input where $t_n$ is the dimension of the t ask label of each task. The input is passed through a fully connected layer of dimension 64 to produce the embedding for each of the task's training samples. The extracted embedding is then passed through a series of 3 attention blocks. Each block is comprised of a dot-product self-attention operation followed by two fully connected layers of dimensions 128 and 64 respectively and finally a layer normalization operation (Ba et al., 2016) at the end. The first fully connected layer after the self-attention module in each block is also followed by a ReLU non-linearity activation function (Nair & Hinton, 2010). Each block also has a residual connection (He et al., 2016) from the embedding to the output of the second fully connected layer. The attention function we used is the scaled dot-product attention introduced by Vaswani et al. (2017):

$$Att(Q, K, V) = softmax(\frac{QK^T}{\sqrt{d_k}})V, \tag{5}$$

where $Q$, $K$ and $V$ represent the Query, Key and Value of the Attention module respectively. $d_k$ represents the dimension of $K$ and is used as a scaling factor. In our case as we using self-attention, $Q$, $K$ and $V$ are all represented by the task embedding. The output of the attention blocks are then passed though another fully connected layer of dimension 40 to ensure the same dimensionality as the sample features. The final weight vector, $w$ is obtained by taking the mean of the output.

## A.3   TASK LABEL GENERATOR

The Task Label Generator takes in a 41 dimensional input has a similar architecture to the Weights Generator. It consists of a 64 dimensional fully connected layer followed by 3 attention blocks similar to that in the Weights Generator. The Task Label Generator outputs a $t_n$ dimensional task label for each training sample and the final task label, $t$ is obtained by taking the mean of all the individual task labels. The generated task label is then passed through a sigmoid activation layer.

## A.4   TRAINING PROCEDURE

We train our model in two stages. In the first stage, only the Feature Extractor and Weights Generator are updated. In this stage we use the true task labels from the regression tasks as input to the Weights Generator. The loss function used is mean squared error between the labels of the validation set labels and the prediction.

$$L(\theta^F, \theta^G) = \sum_{k=j}^{J}(y_j^{val} - y_j')^2 \tag{6}$$

Where $j$ is number of samples in the validation set. As the Feature Extractor is tasked to only map a low dimensional samples $x$ to high dimensional features $f$, through training the Feature Extractor eventually learns to map samples to features that are most suited for the task domain. The Feature Extractor can therefore be thought as implicitly modeling and capturing the task distribution $p(\mathcal{T})$.

For any given regression task, the set of features $f$ is fixed. The reason that the model is able to predict different $y'$ value from the same $f$ is purely due to the weights vector $w$. As the Weights Generator has access to both sample label and task label information, it is trained to output weights vectors that capture a specific task instance $\mathcal{T}_i$ within the task distribution $p(\mathcal{T})$.

In the second stage, we train the entire model including the Task Label Generator. In this stage the predicted task labels from the Task Label Generator are used as input to the Weights Generator instead of the true task labels. We also use a identical mean-squared loss function for the this stage of training but instead updating all three sets of parameters $\theta^F$, $\theta^G$ and $\theta^T$.

## B  DATASET AND TRAINING DETAILS

Here we provide details on all of our regression tasks and some additional training details of our model. For all the regression tasks we normalize the task labels to a $[0, 1]$ range using it for training. We also provide visual results of our model's performance on all three regression tasks in the following section

### B.1  SINUSOIDAL REGRESSION

For the sinusoidal regression task, we follow the experimental set up in Yoon et al. (2018). The target function is defined as $y(x) = Asin(\omega x + b) + \epsilon$, where amplitude $A$, phase $b$, frequency $\omega$ and noise $\epsilon$ are the parameters of the function. We sample the each parameters uniformly from range $A \in [0.1, 5.0]$, $b \in [0, 2\pi]$ and $\omega \in [0.5, 2.0]$. For noise $\epsilon$, we sample it from distribution $N \sim (0, 0.01A^2)$. We use the amplitude, phase and frequency values to form a 3-dimensional task labels for the sinusoidal task

### B.2  1D HEAT EQUATION

For the heat equation task, we define it as such: Consider a 1-dimensional rod of length $L$ with both of its ends connected to heat sinks, *i.e.* the temperature of the ends will always be fixed at $0K$ unless a heat source is applied at the end. a constant point heat source is then applied to a random point $s$ on the rod such the the heat point source will always have a temperature of $1.0K$. We would like the model the temperature $u(x, t)$ at each point of the rod a certain time $t$ after applying the heat source until the temperature achieves equilibrium throughout the rod. The temperature at each point $x$ after time $t$ is given by the heat equation:

$$\frac{\partial u}{\partial t} = k\frac{\partial^2 u}{\partial x^2}$$

In our experiments, we set $L$ to be 5 and randomly sample 10 points of position range $[0, 5]$ on the heat equation curve for a 10-shot regression task. We use both the $x$ and $t$ to form a 2 dimensional task label for the heat equation task.

### B.3  2D GAUSSIAN

For the 2D Gaussian tasks, the model is trained to predict the probability distribution function of a two-dimensional Gaussian distribution. We train our model from Gaussian distribution task with mean ranging from $(-2, -2)$ to $(2, 2)$ and standard deviation of range $[0.1, 2]$. We fix the standard deviation to be of the same value in both directions. The model is once again trained on 10 randomly sampled input points for a 10-shot regression task.We plot the predictions of points in the range of $[\mu - 3, \mu + 3]$ where $\mu$ is mean of the Gaussian task.. Similar to the heat equation tasks, we use the mean and standard deviation values to form a 3 dimensional task label.

### B.4  ADDITIONAL TRAINING DETAILS

For all of our experiments except the one comparing against BMAML, we train both our model on regression tasks of batch size 32 for 50000 steps for both training stages. For experiment comparing against BMAML for the sinusoidal task, we train our model on tasks of batch size 10 for 10000 steps using a pre-trained Task Label Generator trained on 50000 training steps. We also limit the data set in this experiment to just 1000 regression tasks for fair comparison. We use the Adam Optimizer (Kingma & Ba, 2014) as the optimization method to preform stochastic gradient decent on our model. We implement all our models using the Tensorflow (Abadi et al., 2016) library and train them on a Nvidia GTX 1080 Ti GPU.

## C ADDITIONAL ABLATION STUDY

In this section, we further illustrate our justification made for our model by conducting an additional ablation study. We intend to show that that learned features $f$ indeed correspond to a set of basis functions $\{\phi_i(x)\}$ that correspond to the function $F(x)$. If that is indeed the case, each of the learned basis function $\phi_i(x)$ should correspond to certain characteristics of the regression function, and removing one basis function from the final prediction should give noticeable difference in the models final prediction. Thus, we conduct an ablation study where we remove one of the dimensions in $f$ and visualize the change in the final prediction.

We conduct this study on sinusoidal regression task show the results in Figure 2. The results shows that removing certain dimensions from the feature vector does give significantly different results and does correspond to certain characteristics of the regression function. Namely for the first case, the removed feature correspond the accurate magnitude and position of the regression curve whereas for the second case, the removed feature correspond to the "shape" of the regression curve at range [-2.0 ,2.4].

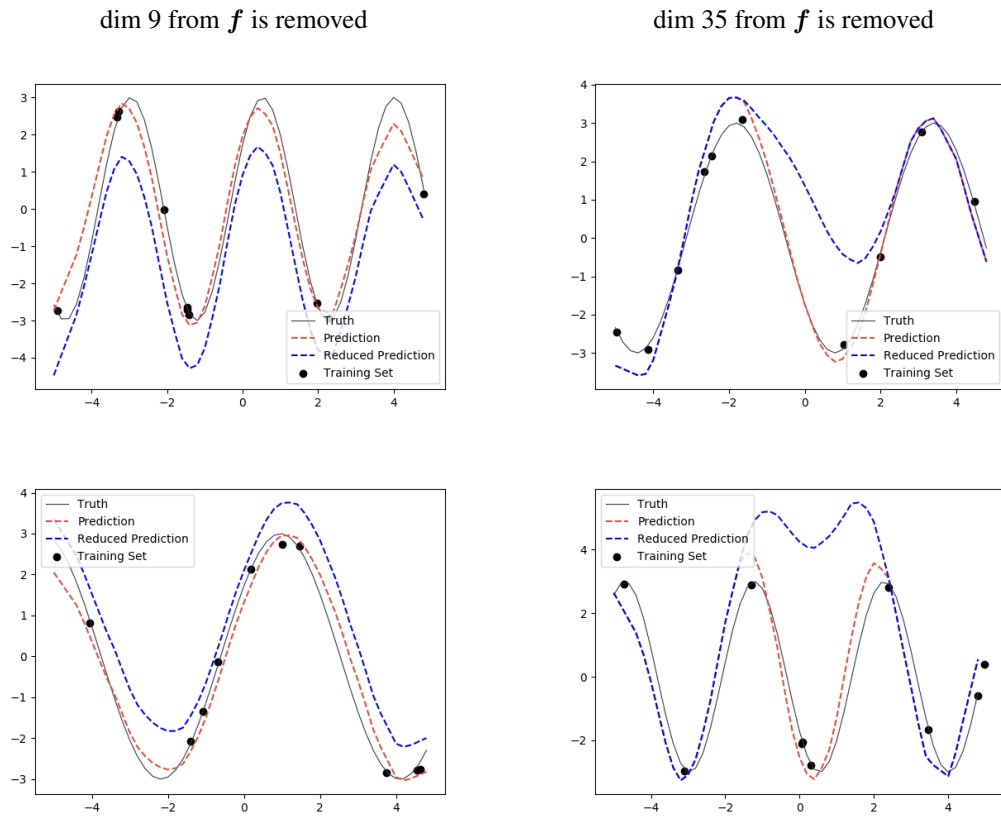

Figure 2: Results of the ablation study where we remove one dimension in $f$ from the final prediction and compare the change to that of the original prediction with all of the dimensions in $f$. The prediction made with reduced dimension is represented by the blue dashed line.

## D VISUAL RESULTS

In this section we show some example visual results of our model's results on the three regression tasks. In Figure 3, we show some examples of our models results on the sinusoidal task. In Figure 4, we show examples of our model results on the 1D heat equation task and compare it against the results from MAML (Finn et al., 2017). Finally in Figure 5, we show examples our model results on the 2D Gaussian task.

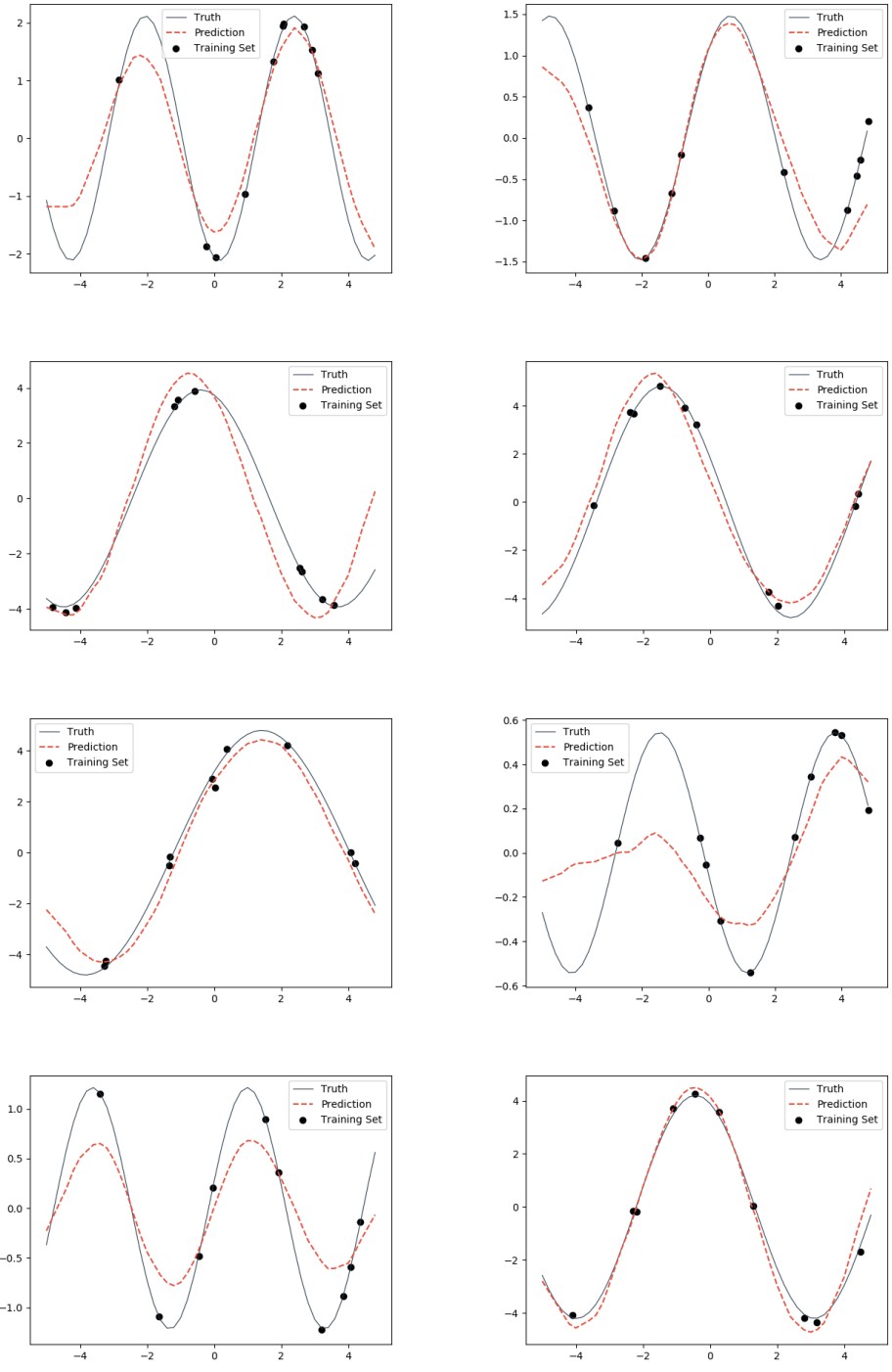

Figure 3: Our model's results on sinusoidal tasks

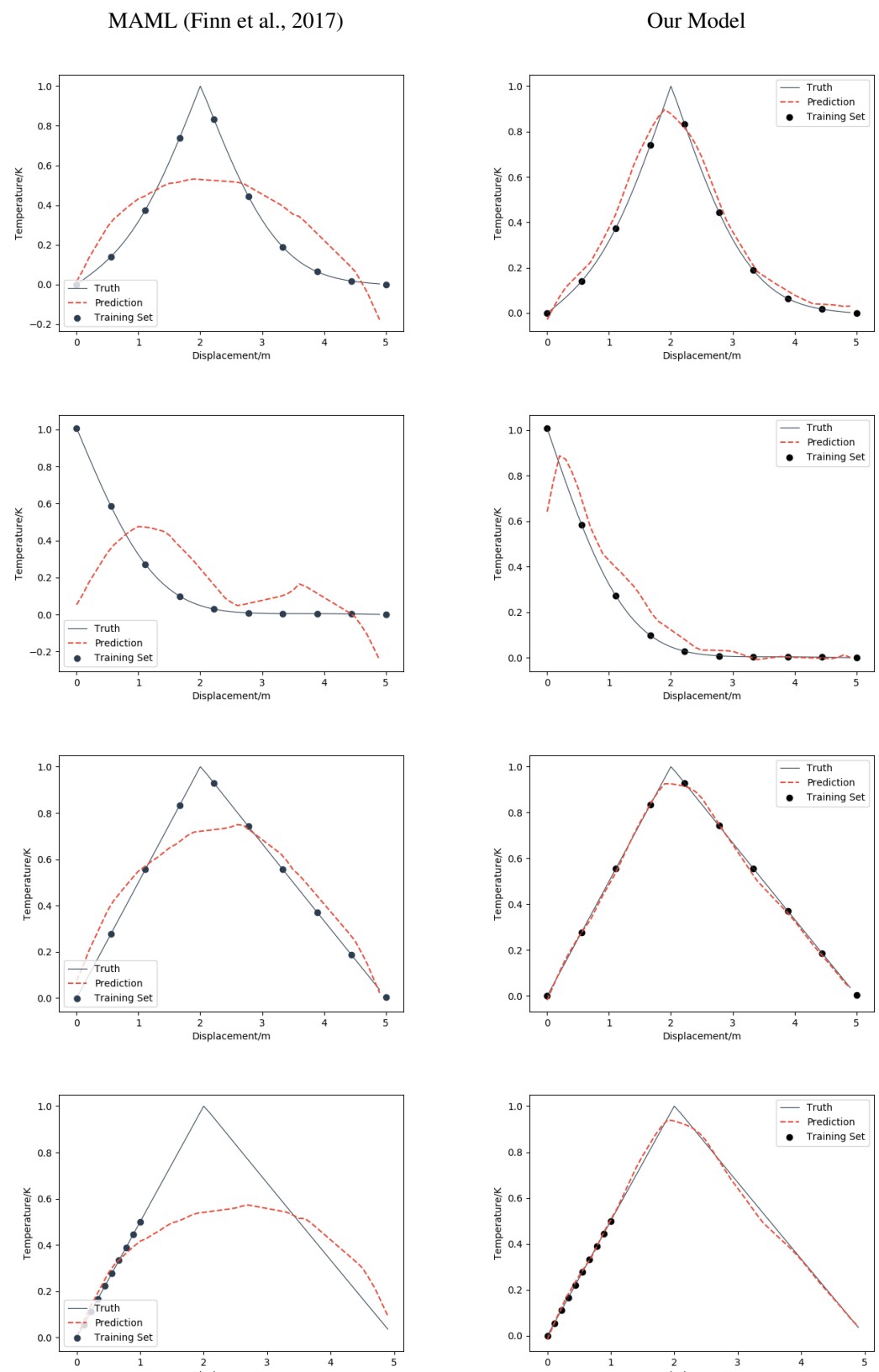

Figure 4: Our models performance on the 1D Heat Equation Task. We also provide a comparison the performace of MAML on the left.

Ground Truth                                            Prediction

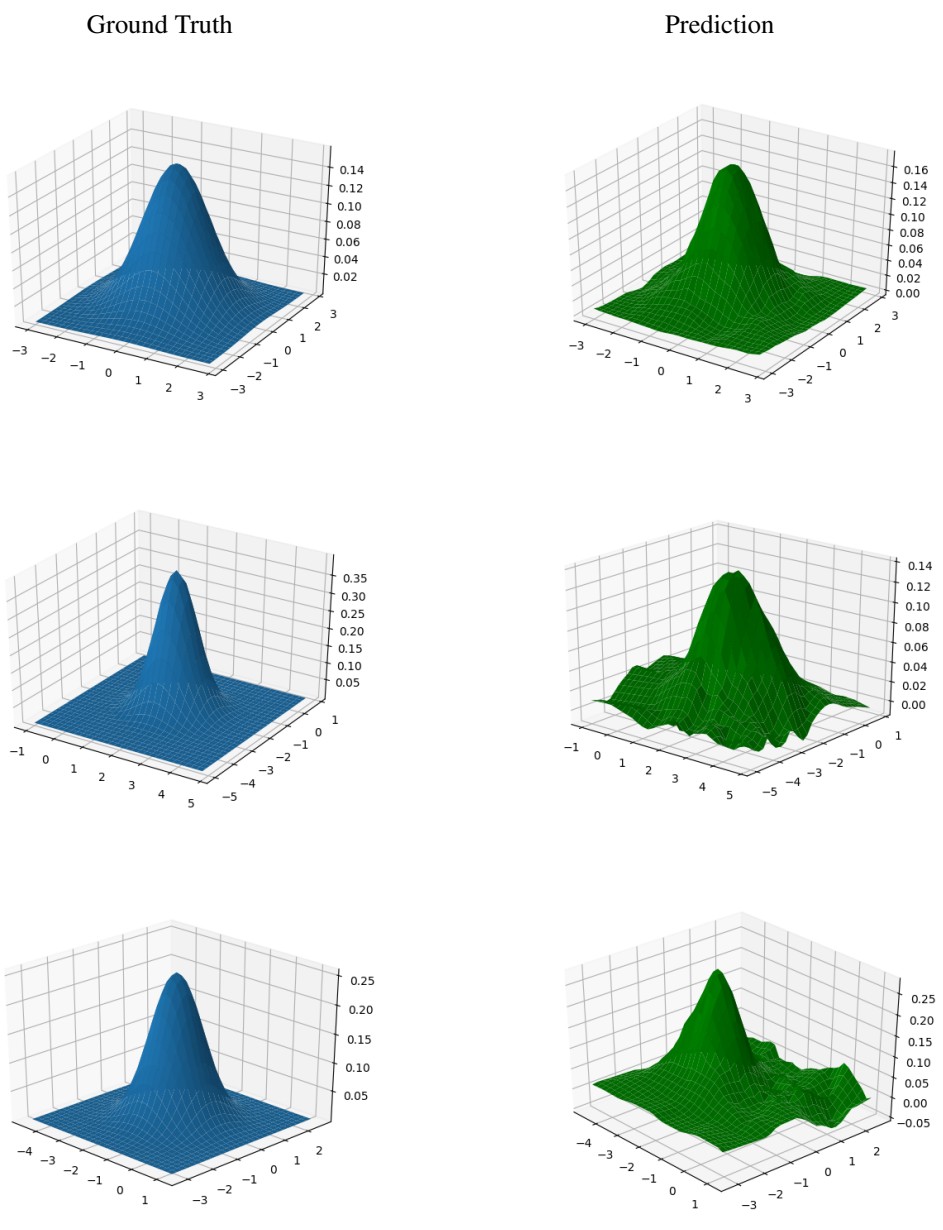

Figure 5: Our Model's results on 2D Gaussian Tasks.

