# OpenReview forum: "Few-Shot Regression via Learned Basis Functions"
_ICLR.cc/2019/Workshop/LLD — LLD 2019_

### Official Review · AnonReviewer2 · 2019-04-07
**This paper proposes an interesting way learn spasre feature representation to address small sample regression problems.**

**Rating:** 3
**Confidence:** 2

**Review:**

This paper proposes a shot-learning method for small sample regression problems. Each regression problem consists of several tasks, each defines an input and output relation. Given samples of different tasks, the goal is to learn these task-dependent relations. The idea is first to learn a sparse feature representations which is task-independent. Then using the feature, output, and task label information to learn a task-specific mapping from the feature to output. These two steps are realized by Feature extractor and Weight generator. When given a new task, one needs to learn the task label of input mapping from the feature to output. This requires to output samples specific to the new task, which is done by task label generator.

The method seems to be novel and it works well on several regression problems. The idea of sparsity seems to be essential to achieve good estimation. It deserves further understanding. Following the result in Table 1 and 2, the Task Label and its generator plays an important role in 1d case. In 2d, the result is less conclusive since the confidence interval is too big to compare tasks with task label generator and no task label generator, not sure if it is due to the task label generator’s error.

Overall, the idea and results are interesting. The effects of adding task label could also be done on the two new regression problems. This would be interesting to discuss as well.

---

### Official Review · AnonReviewer1 · 2019-04-10
**Interesting method proposed**

**Rating:** 3
**Confidence:** 2

**Review:**

The paper proposes a method that learns a regression model with a few samples.

Pros:
- It is an interesting application.
- Original work and clearly explained. Mathematically sound.
- It outperforms other methods.

Cons:
- Just a few examples in the results section. Part of the results were attached as appendices. Looking at the results, my question would be how the different models compared in Tables 1 and 2 perform in the different regression data sets. Only one model is compared for each regression data set.

---

### Decision · Program_Chairs · 2019-04-11
**Acceptance Decision**

Accept